# Generation and Characterization of Stable Redox-Reporter Mammalian Cell Lines of Biotechnological Relevance

**DOI:** 10.3390/s22041324

**Published:** 2022-02-09

**Authors:** Karen Perelmuter, Inés Tiscornia, Marcelo A. Comini, Mariela Bollati-Fogolín

**Affiliations:** 1Cell Biology Unit, Institut Pasteur de Montevideo, Mataojo 2020, Montevideo 11400, Uruguay; kperelmuter@pasteur.edu.uy (K.P.); tiscornia@ort.edu.uy (I.T.); 2Laboratory Redox Biology of Trypanosomes, Institut Pasteur de Montevideo, Mataojo 2020, Montevideo 11400, Uruguay

**Keywords:** rxYFP, biosensor, GM-CSF, bioprocess development, redox metabolism

## Abstract

Cellular functions such as DNA replication and protein translation are influenced by changes in the intracellular redox milieu. Exogenous (i.e., nutrients, deterioration of media components, xenobiotics) and endogenous factors (i.e., metabolism, growth) may alter the redox homeostasis of cells. Thus, monitoring redox changes in real time and in situ is deemed essential for optimizing the production of recombinant proteins. Recently, different redox-sensitive variants of green fluorescent proteins (e.g., rxYFP, roGFP2, and rxmRuby2) have been engineered and proved suitable to detect, in a non-invasive manner, perturbations in the pool of reduced and oxidized glutathione, the major low molecular mass thiol in mammals. In this study, we validate the use of cytosolic rxYFP on two cell lines widely used in biomanufacturing processes, namely, CHO-K1 cells expressing the human granulocyte macrophage colony-stimulating factor (hGM-CSF) and HEK-293. Flow cytometry was selected as the read-out technique for rxYFP signal given its high-throughput and statistical robustness. Growth kinetics and cellular metabolism (glucose consumption, lactate and ammonia production) of the redox reporter cells were comparable to those of the parental cell lines. The hGM-CSF production was not affected by the expression of the biosensor. The redox reporter cell lines showed a sensitive and reversible response to different redox stimuli (reducing and oxidant reagents). Under batch culture conditions, a significant and progressive oxidation of the biosensor occurred when CHO-K1-hGM-CSF cells entered the late-log phase. Medium replenishment restored, albeit partially, the intracellular redox homeostasis. Our study highlights the utility of genetically encoded redox biosensors to guide metabolic engineering or intervention strategies aimed at optimizing cell viability, growth, and productivity.

## 1. Introduction

Mammalian cell lines have become the dominant system for the production of recombinant proteins for clinical applications, mainly due to their ability to produce diverse, correctly folded, and glycosylated proteins [1]. Compared to other cell systems, protein production by mammalian cells is limited because of the low yields and low processing rate. However, over the last 20 years an increase in process yields from 0.5 to 9 g/L was achieved [2]. Moreover, innovations in the manufacturing processes also aim to achieve product stability and quality along with low manufacturing costs, in reduced times [3].

In order to optimize the production process, several parameters such as cell growth and division, metabolic activity, and physiological stress have been evaluated. Interestingly, the most important stress observed in cells in culture is oxidative stress [4]. In biopharmaceutical production, oxidative stress produced under hypoxic and hyperoxic conditions is associated with an increase in the metabolic flux towards the oxidative phosphorylation pathway. Such metabolic rewiring satisfies the elevated energetic demands, but lead to the accumulation of metabolic byproducts and proteins in the endoplasmic reticulum, especially in (fed) batch cultures ([4] and references therein). Moreover, mitochondrial physiology has been rated as a critical parameter determining culture performance of hybridoma cell cultures [5,6].

Multiple metabolic processes as well as signaling and transcription are tightly regulated by the cellular redox state. In this regard, protein-bound and free thiols are key players in redox sensing and regulation of protein function. All eukaryotes are endowed with thiol-redox systems (e.g., Thioredoxin and Glutaredoxin systems) that, in a compartment-specific fashion, contribute to maintaining a physiological intracellular thiol-redox balance [7]. Both systems depend on the reducing power of NADPH, which *via* a multicomponent enzymatic cascade is funneled towards different effector proteins (e.g., thiol/disulfide oxidoreductases, peroxidases, ribonucleotide reductase, etc.) or the thiol-containing tripeptide glutathione (GSH). GSH is the most abundant low molecular mass thiol in eukaryotic cells. It acts as redox cofactor for glutaredoxins (Grx) and peroxidases, which catalyze the reversible thiolation of protein cysteines and remove different types of hydroperoxides, respectively [8,9,10]. Both functions serve protective and regulatory roles in cells [8]. Under optimal growth conditions, the cytosol of eukaryotic cells has a very high ratio of reduced vs. oxidized glutathione (GSH/GSSG ratio) [8,9,10,11,12]. Thus, any perturbation of this ratio, either by impaired biosynthesis of GSH (an ATP-dependent process) and/or consumption (e.g., oxidation, conjugation or extrusion), is usually an indicator of cellular (oxidative) stress at cytosolic level [13].

The GSH/GSSG ratio has been shown not to be uniform inside cells but to vary in a compartment-specific fashion [12,14,15]. Therefore, the in situ measurement of the redox state of the GSH pool by non-invasive methods is considered relevant to achieve an accurate diagnosis of the cell metabolic state [16]. In this regard, several laboratories have developed redox-sensitive variants of different fluorescent proteins [17,18,19,20]. All of them share a common biosensing mechanism: two spatially proximal and surface-exposed cysteine residues (introduced by genetic engineering) located in regions containing amino acids that interact with the protein chromophore are reversibly oxidized by GSSG in a reaction catalyzed by Grx; the concomitant formation of a protein cysteine disulfide leads to discrete and allosteric conformational changes near the chromophore that modify its spectral properties (i.e., decrease in fluorescence absorption and emission at the maximum peak); the cysteine disulfide can be reduced by GSH/Grx, which restores the chromophore properties to its ground state (i.e., increase in fluorescence absorption and emission at the maximum peak) [21,22]. Most redox biosensors are intensiometric, *ergo*, they present a single absorption/emission peak. The exception being roGFP and congeners [18] and a mutant version of Clover protein [23], which have a ratiometric behavior with two absorption/emission peaks that respond in an opposite fashion to the redox changes. By linking Grx, peroxidases, sulfur-transferases or thioredoxin to the fluorescent protein module, the kinetic response and/or specificity of the biosensor for different redox metabolites (e.g., GSH/GSSG, H_2_O_2_, mercaptopyruvate, thioredoxin, thiosulfate) can be fine-tuned [16,21,22,24,25].

These genetically encoded biosensors overcome the limitations of conventional redox measurements based on chemical compounds that often lack compartment-specific localization, specificity and reversibility, are toxic and prone to undergoing side (redox) reactions [16,22,26,27,28,29]. Since its development, two decades ago, the fluorescent protein redox biosensors have been extremely useful for answering and exploring key concepts of cell and organismal redox biochemistry and biology in health and disease (reviewed in [21,22]). 

Chinese hamster ovary (CHO) and human embryo kidney 293 (HEK-293) lead the ranking of mammalian hosts used for the production of therapeutic proteins [1]. Indeed, about 70% of the marketed biopharmaceutical proteins, and almost all mAbs, were produced in CHO cells [30,31]. Their popularity lies in their high productivity, consistent growth phenotypes, suitability for large-scale industrial culturing, ease of adaptation to various chemically defined media, lower susceptibility to infection by human viruses, and capability to perform human compatible glycosylation [3]. The HEK-293 cell line has been extensively used for the production of viral vectors for gene and cell therapy, and has also been validated as an efficient platform for the large-scale production of recombinant proteins due to its high transfectivity, rapid growth rate, and ability to grow in a serum-free, suspension culture [32,33]. Despite the impact of redox metabolism in cellular productivity, so far, there are no studies reporting the use of genetically encoded biosensors to monitor the intracellular redox environment in a bioprocess framework.

In the present study, HEK-293 cells and a CHO cell line that constitutively express the human granulocyte macrophage colony stimulating factor (hGM-CSF) were selected to investigate the fluctuations in their cytosolic GSH/GSSG pool under physiological conditions. GM-CSF is an early acting factor essential for the regulation and differentiation of hematopoietic progenitor cells as well as for stimulating the activation of mature cell populations [34]. This lymphokine is used to treat myeloid leukemia and to stimulate the granulocyte and macrophage production in patients suffering from innate or acquired (e.g., viral, radiation, chemotherapy) immunodeficiency [35].

Here, we report the generation and characterization of two transgenic cell lines that express stably in their cytosol a redox biosensor based on the yellow fluorescent protein (rxYFP). The growth and metabolic phenotype of the redox reporter cells was compared to the parental cell lines. The sensitivity of the intracellular biosensor to respond to a physiological oxidant, namely hydrogen peroxide, was investigated. The characterization of the CHO-hGM-CSF redox reporter cell line included metabolic-bioprofiling and -complementation studies in a batch culture system. The biosensor proved valuable in anticipating and linking changes in the GSH/GSSG pool to cell metabolic reprogramming.

## 2. Materials and Methods

### 2.1. Reagents

Unless otherwise indicated, all chemicals used were of highest grade and purchased from Sigma Aldrich (St. Louis, MO, USA). Media, fetal bovine serum (FBS) and consumables for cell culture were obtained from Life Technologies (Carlsbad, CA, USA), GE Healthcare (Chicago, IL, USA) and Greiner (Kremsmünster, Austria).

### 2.2. Cell Culture

Human embryonic kidney 293 (HEK-293, ATCC^®^ CRL-1573™) and CHO-K1-hGM-CSF [36] cells were cultured in DMEM supplemented with 10% FBS. Cells were routinely propagated in 25 or 75 cm^2^ tissue culture flasks at 37 °C, 5% CO_2_ in a humidified incubator until reaching 60–70% confluence. Unless otherwise stated, all assays were initiated with cells from exponentially growing cultures as described next. Cells were trypsinized and cultured in fresh medium 24 h prior to the assay and, further trypsinized and seed in fresh medium at the required cell density. Cells used for assays were cultured for less than 15 passages. 

### 2.3. Growth and Metabolite Analysis

Batch cultures were performed in 12-well plates. For CHO-K1-hGM-CSF, 5 × 10^4^ cells/well and for HEK-293 5 × 10^5^ cells/well were seeded in DMEM medium supplemented with 10% FBS (1 mL/well). Total cell number and viability were determined daily using the trypan blue exclusion method and counted in a hemocytometer. Culture supernatants were collected every day, clarified, and glucose and lactate were determined using the Glucose/Lactate analyzer (BioProfile Basic 2, Nova Biomedical, Waltham, MA, USA). Ammonia was quantified according to [37] with modifications. Amino acids, except for tryptophan, threonine, cysteine, asparagine, glutamine, were quantified using the ACCQ-Fluo Reagent Kit (Waters, Milford, MA, USA). Briefly, amino acids were derivatized with 6-aminoquinolyl-N-hydroxysuccinimidyl carbamate and then separated and detected with a fluorometric HPLC.

### 2.4. hGM-CSF Activity

hGM-CSF biological activity was determined using a cell-based assay. The hGM-CSF-dependent proliferation of TF-1 cells (ATCC CRL-2003) was quantified using MTT (3-(4,5-Dimethylthiazol-2-yl)-2,5-Diphenyltetrazolium Bromide). Routinely, TF-1 cells were cultured in RPMI medium supplemented with 10% FBS and 20 ng/mL rhGM-CSF (Peprotech) at 37 °C, 5% CO_2_ in a humidified incubator. For the assay, exponentially growing cells were collected and washed with RPMI medium. Then, 2 × 10^4^ cells/well were seeded in a 96-well microplate and incubated for 48 h in the absence (negative control) or presence of various concentrations of rhGM-CSF (standard curve) and serial dilutions of CHO-K1-hGM-CSF culture supernatant. MTT solution (0.5 mg/mL) was added and incubated for 3 h, then, SDS (20%) was added to dissolve the precipitate. The soluble product was measured using a microplate reader (Thermo Fisher Scientific, Waltham, MA USA) and a wavelength of 570 nm. 

### 2.5. Generation of Expression Vectors

The plasmid pcDNA3.1Neo carrying the rxYFP-gene was kindly provided by Dr. G. Pani (Istituto di Fisica, Universitá Cattolica S. Cuore, Milan, Italy). This plasmid was used for the transfection of HEK-293 cells. For CHO-K1-hGM-CSF cells, the rxYFP gene was first inserted into the HindIII and XhoI restriction sites of pcDNA3.1 Zeo, using standard molecular biology techniques. 

### 2.6. Generation of Stable Redox Reporter Cell Lines

For the generation of the transgenic reporter cell lines, 1 × 10^6^ cells were seeded in a 25 cm^2^ T-flask and transfected using 10 µg of each plasmid and Lipofectamine^®^ 2000 according to the manufacturer’s instructions. Resistant colonies were selected and expanded in medium containing 400 μg/mL zeocin (for CHO-K1-hGM-CSF) or 400 μg/mL geneticin (for HEK-293) for 30 days. Before sorting, the cell suspension was filtered by a 50 μm nylon mesh and single resistant cells separated using a MoFlo XDP cell sorter (Beckman Coulter, Brea, CA, USA) in “single cell” mode with 0.5 drop sort envelope criteria. Cells were distributed into individual wells of a 96-well plate containing 100 μL culture medium supplemented with penicillin/streptomycin. rxYFP was excited at 488 nm (Argon laser) and fluorescence emission detected by a band-pass filter of 530/40 nm. The sort decision was made based on the analysis of a Forward Scatter (FSC) *versus* Side Scatter (SSC) dot plot, excluding doublets and gating rxYFP cells in a FSC *versus* rxYFP fluorescence dot plot. Clones that grew after the single cell deposition method were amplified, cryopreserved and characterized by flow cytometry and microscopy.

### 2.7. Flow Cytometry Analysis

Flow cytometry data acquisition and analysis was performed using a CyAn™ ADP Flow Cytometer (Beckman Coulter, Brea, CA, USA) and the Summit 4.3 Software, respectively. rxYFP and Propidium Iodide (PI) fluorescence was detected by exciting at 488 nm and collecting emission with band-pass filters of 530/40 and 613/20 nm, respectively. For each sample, a minimum of 10,000 counts gated on a FSC *versus* SSC dot plot and excluding doublets were recorded. The rxYFP mean fluorescence intensity (MFI) was analyzed only for the population of viable cells (i.e., PI negative cell population). 

### 2.8. Confocal Microscopy

Cells in the exponential growth phase were seeded in sterile glass bottom dishes (Thermo Fisher Scientific, Waltham, MA, USA) and incubated for 24 h at 37 °C, 5% CO_2_ in a humidified incubator. Before analysis, the culture medium was replaced with fresh medium and the cells were challenged with H_2_O_2_ 2.5 mM and/or DTT 5 mM. Multiposition time-lapse imaging with a 64X oil immersion objective (NA 1.42) and Z-stacks of 20–30 μm (step size of 1–1.5 μm) was performed at 37 °C and 5% CO_2_ using an inverted confocal laser microscope (Leica TCS SP5). The signal for rxYFP was detected using the following excitation and emission wavelengths: λex/λem 488/565 ± 35nm. Images were acquired with the LAS AF Lite software (Leica Microsystems, Wetzlar, Germany) and further analyzed with the ImageJ processing program.

### 2.9. Physiological Assays

#### 2.9.1. Dose–Response Curve for H_2_O_2_

CHO-K1-hGM-CSF and HEK-293 cells expressing the rxYFP biosensor were treated for 30 min at 37 °C with H_2_O_2_ (ranging from 1 µM to 2.5 mM) and then rxYFP fluorescence of viable cells was evaluated by flow cytometry (Section 2.7). The percentage of rxYFP reduction was calculated as indicated in Section 2.10.

#### 2.9.2. Treatment with rhGM-CSF

CHO-K1-hGM-CSF cells expressing rxYFP were seeded at 1.5 × 10^4^ cells/cm^2^ and, after 24 h of incubation, the medium was renewed with or without the addition of rhGM-CSF (1 µg/mL). Cells were further incubated for 24 h, trypsinized and analyzed by flow cytometry (Section 2.7 and Section 2.10).

#### 2.9.3. Glucose and FBS Deprivation

CHO-K1-hGM-CSF cells expressing rxYFP were seeded at 3.3 × 10^4^ cells/cm^2^ and incubated for 24 h in fresh culture medium. Next, the medium was removed, the cell monolayer washed with PBS, and fresh medium without glucose or SFB was added. After 8 or 24 h incubation, cells were trypsinized and analyzed by flow cytometry (Section 2.7 and Section 2.10). 

### 2.10. Redox Sensor Calibration

For calibration purposes, all studies involving the monitoring of intracellular rxYFP redox state included cell samples treated with 2.5 mM H_2_O_2_ for 30 min or DTT (at 1 and 10 mM for CHO-K1-hGM-CSF and HEK-293, respectively) for 20 min. After treatments, MFI was determined by flow cytometry as described above. The percentage of biosensor reduction (% rxYFP reduction) was calculated as follows:$$\%\;\mathrm{rxYFP}=100\;\%\;\times \left(\frac{\mathrm{MFI}\;\mathrm{sample}-\mathrm{MFI}\;H_{2}O_{2}}{\mathrm{MFI}\;\mathrm{DTT}\;-\mathrm{MFI}\;H_{2}O_{2}}\right),$$
where MFI sample, MFI H_2_O_2_, and MFI DTT correspond to the mean fluorescence intensity of rxYFP in the sample under study, in the fully oxidized condition (H_2_O_2_-treated cells), and in the fully reduced condition (DTT-treated cells), respectively.

### 2.11. Statistical Analysis

Data are expressed as the mean +/− standard deviation of technical triplicates from three independent experiments. Statistic calculations were performed using the GraphPad Prism Software version 5.00 Demo (San Diego, CA, USA). Differences were considered statistically significant when *p* < 0.05 using an unpaired Student t-test.

## 3. Results

### 3.1. Generation of Transgenic Cell Lines Expressing the Redox Biosensor

Among the different fluorescent protein-based redox biosensors suitable for monitoring intracellular redox changes, we opted for rxYFP and roGFP2 given that they present high quantum yields and most flow cytometry devices are equipped with the corresponding laser/filter to detect their fluorescence.

Our attempts to generate stable cell lines expressing the chimeric fusion hGrx-roGFP2 proved unsuccessful since the transfected cells did not survive the selection with the antibiotic. In contrast, stable cell lines of CHO-K1-hGM-CSF and HEK-293 expressing the rxYFP protein could be obtained after 30 days of antibiotic selection. Individual clones displaying different expression levels of rxYFP were separated by cell sorting and single cell deposition. A preliminary phenotypic characterization of randomly chosen clones (n = 5) showed that they share similar growth and metabolic behaviors as well as response to oxidant stimuli (data not shown). Therefore, only one clone from each cell line (C7 for CHO-K1-hGM-CSF, hereafter called CHO-GM-rxYFP and B6P2 for HEK-293, hereafter called HEK-293-rxYFP) was subjected to further characterization. 

### 3.2. The Redox Reporter Cell Lines Respond in a Dynamic and Sensitive Fashion to Redox Stimuli 

In order to verify the stable integration and expression of rxYFP, CHO-GM-rxYFP and HEK-293-rxYFP cell lines were grown for three months in culture medium lacking the selective antibiotic and analyzed by flow cytometry. The analysis showed that more than 80% of the cell population from both reporter cell lines maintained a homogenous and high expression of the redox biosensor (data not shown).

The intracellular response of the biosensor to an oxidant and reducing stimuli was tested in both reporter cell lines grown at exponential phase. The MFI of rxYFP for cells treated with 1 mM DTT (20 min), a permeant reducing agent, was almost identical to that of untreated cells (Figure 1A,B), indicating that the biosensor was fully reduced in exponentially growing cells. In contrast, a 30 min incubation with 2.5 mM of the physiological oxidant H_2_O_2_ led to a 51% and 61% decrease in the MFI for CHO-GM-rxYFP and for HEK-293-rxYFP, respectively. Importantly, incubation with DTT (1 mM for 20 min) of H_2_O_2_-treated cells induced a 66% and 90% recovery in the MFI of CHO-GM-rxYFP and HEK-293-rxYFP cells, respectively. Overall, these results confirm the capacity of the redox biosensor to respond in a dynamic fashion to redox stimuli.

The response of the redox biosensor in CHO-GM-rxYFP cells was also monitored by confocal microscopy (Figure 1C,D). Time-lapse measurement of rxYFP signal after H_2_O_2_ treatment (2.5 mM; upper panel Figure 1C) showed a sustained decrease in fluorescence intensity that was reverted upon treatment with DTT (1 mM; lower panel Figure 1C). For 10 representative cells, the relative fluorescence was calculated as the MFI normalized to the cell area and the MFI at time 0 (Figure 1D). Despite some cell-to-cell variations in relative fluorescence, the change in fluorescence showed a similar trend upon the oxidative and reductive challenges. 

Notably, only a minor fraction (3%) of the redox biosensor was found in an oxidized state in exponentially growing cultures from both reporter cell lines, which indicates a highly reducing cytosolic environment. 

Next, the capacity of the reporter cell lines to respond in a concentration-dependent fashion to a physiological oxidant, namely H_2_O_2_, was tested (Figure 2). In both, CHO-GM-rxYFP and HEK-293-rxYFP cells, the degree of biosensor oxidation was proportional to the concentration of H_2_O_2_. However, the magnitude of biosensor oxidation differed between both cell lines. For instance, CHO-GM-rxYFP cells proved comparatively more sensitive to detecting H_2_O_2_ concentrations within a physiological range (1–100 μM) than HEK-293-rxYFP cells. In fact, CHO-GM-rxYFP cells reacted to as low as 1 μM H_2_O_2_ (~15% oxidation of rxYFP) and at 10, 50, and 100 μM H_2_O_2_, the level of biosensor reduction in these cells (65%, 37%, and 30%, respectively) almost halved that achieved in HEK-293 cells (95%, 75%, and 65%, respectively). At concentrations above 100 μM H_2_O_2_, biosensor oxidation (≥80%) is remarkable and did not differ significantly between both reporter cell lines. Concentration–response assays were performed for other CHO-GM-rxYFP clones expressing lower levels of rxYFP (evidenced by lower MFI) and an overall similar sensitivity towards H_2_O_2_ was observed (data not shown). This suggests that factors others than the level of rxYFP expression account for the higher H_2_O_2_-sensitivity of CHO-GM *versus* HEK-293 cells.

### 3.3. The Expression of the Redox Biosensor Does Not Affect the Phenotype of the Host Cells

A comparative analysis of different growth and metabolic parameters was performed for CHO-GM-rxYFP and HEK-293-rxYFP cells and their corresponding parental (non-reporter) cell lines. Cell density and viability, glucose, lactate, ammonia, and rhGM-CSF activity (only for CHO-GM-rxYFP) were determined daily in batch cultures from each cell line (Figure 3). 

The growth rate, the total number, and percentage of viable cells (≥90%) at the different culture phases were almost identical for the redox reporter and the parental cell line of CHO-GM (Figure 3A) and HEK-293 cells (Figure 3C). In the case of HEK-293 cells, the percentage of viable cells and cell density was always slightly higher for the reporter cell line than for the parental cell line (Figure 3C). Although CHO-GM-rxYFP cells appear to produce rhGM-CSF earlier (day 2, log phase) than the parental cell line (day 4 on), at stationary phase, both cell lines accumulate a similar amount of bioactive product (Figure 3A).

Overall, the trend of glucose consumption and lactate, and ammonia secretion was indistinguishable between the redox reporter and the parental CHO-GM and HEK-293 cell lines (Figure 3B,D). Nonetheless, both cell types display remarkable metabolic differences. 

For CHO-GM cells, it was interesting to observe that when they nearly stopped their exponential growth (day 3, Figure 3A), glucose level dropped to half the initial concentration in the culture medium, which was mirrored by an equimolar increase in secreted lactate (~12.5 mM; Figure 3B). At the end of the batch culture (day 8), the total amount of glucose (5 mM) and lactate in the culture medium was similar in the redox reporter and parental cell line (30 mM), with the last metabolite being slightly higher than the initial amount of glucose (25 mM). Interestingly, the amount of ammonia secreted to the culture medium was almost negligible during the exponential growth phase (day 1 to 3) of both cell lines but increased steadily from day 3 onwards (from 1–1.2 to 2–2.7 mM at day 8; Figure 3B). This event coincides with glucose becoming a limiting energetic substrate and suggests a complementary increase in amino acid catabolism. 

For the HEK-293 cultures (redox reporter and parental cell lines), glucose was rapidly depleted from the medium from at day 5 and extracellular lactate showed a steady accumulation over time until day 3, the time point at which it achieved a *plateau* concentration (Figure 3D). In the HEK-293 cultures, the equimolar conversion of glucose into lactate preceded by one day that observed in CHO-GM cells and the entrance into stationary growth phase. However, similar to CHO-GM cells, the total amount of lactate secreted (30 mM) approaches that of glucose in fresh medium (25 mM), which suggests that an important fraction of the pyruvate formed during glycolysis is further metabolized by the Krebs cycle. From day 1 onwards, ammonia accumulated steadily in the extracellular medium of HEK-293 cell cultures (Figure 3D) reaching concentrations (≥3.5 mM) higher than those recorded for CHO-GM cells (Figure 3B). 

In order to compare the metabolic performance of the cell lines, the *per* cell rates of growth and metabolite consumption/production during the exponential growth phase (from day 0 to day 2) was estimated (Figure 4). Overall, the parental and reporter cell lines from each cell type presented very similar metabolic and growth rates. For CHO-GM cells, the only parameter that statistically differed (*p* < 0.05) between the reporter and parental cell line was the rate of lactate production (*p* = 0.03), which was 30% lower in cells expressing the redox biosensor. While for HEK-293 cells, the rate of ammonia formation was significantly higher (40%, *p* = 0.03) in cells expressing rxYFP. 

Independently of the expression of the biosensor, CHO-GM and HEK-293 cells displayed a significantly different glucose catabolism with the hamster-derived cell line showing a higher rate of glucose consumption (*p* = 0.03 and *p* = 0.02 for parental and reporter cell lines, respectively) and lactate secretion (*p* = 0.001 and *p* = 0.002 for parental and reporter cell lines, respectively) than the cell line of human origin. 

In summary, the expression of the redox biosensor does not affect the metabolic phenotype of both cell lines.

### 3.4. Monitoring Intracellular Redox Changes in CHO-GM-rxYFP Batch Cultures 

With the aim of gaining insights into potential changes occurring in the cytosolic GSH/GSSG pool during physiological growth conditions, the level of biosensor reduction was monitored during a batch culture of CHO-GM-rxYFP cells (Figure 5A). During the first two days of culture (i.e., exponential growth phase), the biosensor showed maximum reduction levels (~100%), indicating a high GSH/GSSG ratio. However, upon day 3 (90% reduction) and onwards, the percentage of reduced rxYFP decreased steadily to reach a minimum of 68% at day 5. The steady-state oxidation level of rxYFP at days 3, 4 and 5 is similar to that caused by concentrations of H_2_O_2_ ranging from ≤1 to 10 μM (Figure 2). Interestingly, the time point at which cells show the first clear signs of an intracellular oxidation of rxYFP, matches their entrance into the stationary phase (Figure 3A). As shown above (Figure 3B), at day 3 cells were metabolically active, as observed by their capacity to metabolize the remaining glucose of the extracellular medium (concentration 12.5 mM). Thus, for a fraction of these cells and on days 3 and 4, the culture medium was replaced by a fresh one and the redox state of rxYFP assessed 24 h later (grey bars in Figure 5A). This procedure halted completely (day 4) or partially (day 5) the level of biosensor oxidation detected at day 3 but was not sufficient to fully restore the highly reducing *millieu* of exponentially growing cells (day 1 and 2). This suggests that at day 3 most cells underwent a metabolic reprogramming that impairs the intracellular GSH/GSSG ratio and cannot be reverted by replenishment of the culture medium. 

Several experiments were conducted to disclose the factor responsible for inducing a more oxidative environment in the cytosol of CHO-K1-hGM-CSF from day 3 onwards. Given that the remarkable biosensor oxidation observed at day 5 of the batch culture (20% oxidation of rxYFP) was partially reverted by culture medium replacement (Figure 5A), the effect of a conditioned medium on the rxYFP redox state was tested in exponentially growing cells. A 24 h incubation of cells with conditioned medium from a 5-day culture did not affect the redox state of rxYFP (Figure 5B). 

hGM-CSF has been reported to trigger ROS formation in hematopoietic cells [38]. Therefore, to determine whether the cytokine may have a similar effect on CHO cells, the reporter cell line was incubated for 24 h with hGM-CSF and the redox state of rxYFP was evaluated. As shown in Figure 5B, extracellular hGM-CSF does not induce oxidation of the intracellular biosensor in CHO-GM cells. 

Furthermore, the possible effect of glucose and FBS deprivation on the redox state of rxYFP was also evaluated. Compared to cells grown in complete culture medium, the lack of glucose or FBS did not significantly affect the intracellular redox state of rxYFP either during short (8 h) or long (24 h) incubation times (Figure 5C).

In order to gain insights into metabolic changes occurring in CHO-GM-rxYFP cells, the amino acid content in the medium supernatant was determined over the batch culture. As shown in Figure 5D, serine, glutamate, and proline were significantly consumed throughout the 6 days culture period, whereas the content of glycine and, in particular, of alanine increased (3- and 6-fold, respectively), indicating an exacerbated formation of them. The concentrations of all other tested amino acids did not vary significantly (±0.5-fold) when compared to their time 0 values. While the production of glycine and alanine showed an almost steady increase during the culture time, glutamate, serine, and proline were rapidly consumed during the exponential and early stationary phase (up to day 4). In particular, glutamate was fully depleted on day 3, which correlates with cell growth arrest (Figure 3A). Worth noting, almost identical growth phase-dependent changes in all these five amino acids were reported previously for CHO cells expressing a recombinant antibody [39].

## 4. Discussion

In the context of animal cell production, the monitoring of the metabolic state of production cell lines is crucial for bioprocess control and development. Classical methods include the quantification of extracellular metabolites in culture supernatant or intracellular metabolites. The use of genetically encoded biosensors offers the possibility to monitor the cell´s metabolic state (both, on a single cell- or population-based fashion) in real-time and non-invasively. In this work, two recombinant cell lines expressing the redox sensor rxYFP, which has specificity to equilibrate with the physiological redox pair GSH/GSSG, were generated and characterized. To our knowledge, the present study is the first reporting the generation of stable redox reporter cell lines of biotechnological interest and on evaluating the intracellular redox changes occurring during a batch culture of a human cytokine-producing cell line.

We choose to express and monitor the GSH/GSSG ratio in the cell cytosol because it is considered to be the milieu for regulation of the redox status of several organelles and compartments with high oxidative activity (i.e., ER, mitochondria, lysosomes, peroxysomes) [40]. 

Importantly, the stable expression of rxYFP did not alter the growth or the metabolic parameters of HEK-293 and CHO-GM-CSF cell lines and shows that, in the exponential phase, the cytosol of both cell types is a highly reducing environment. This observation is in line with the finding of a study performed in HeLa cells where rxYFP redox state was assessed by (redox) Western blotting (non-reducing SDS polyacrylamide gel electrophoresis) [41]. By using non-invasive approaches (flow cytometry or microscopy), here we showed that rxYFP is able to detect small perturbations in the intracellular GSH/GSSG ratio of HEK-293 and CHO-GM cells triggered by H_2_O_2_. In fact, the concentration of H_2_O_2_ attained intracellularly has been estimated to be 650-fold lower than that applied extracellularly due to diffusion and other molecular constraints [42]. Assuming a similar H_2_O_2_ permeability for CHO-GM and HEK-293, this implies that the redox reporter cells were able to detect intracellular variations in GSH/GSSG caused by less than 1 and 10 μM H_2_O_2_, respectively, in a 30 min time framework. Within these concentrations, H_2_O_2_ acts as a signaling rather than an oxidative stress molecule [43].

Notably, CHO-GM cells were more sensitive to the redox unbalance generated by low micromolar concentrations of H_2_O_2_ than HEK-293 cells. Some important factors that may influence the capacity of rxYFP to respond to molecules that perturb the GSH/GSSG ratio are the expression level of the biosensor, the availability of endogenous Grx, the content of cellular antioxidant systems and the cell permeability to the oxidant. For CHO-GM cells, the expression level of rxYFP could not account for the observed higher sensitivity to H_2_O_2_ because clones presenting different contents of the biosensor behaved similarly. Instead, the observed higher capacity of this cell line to respond to H_2_O_2_ may be explained by the high GSH content and upregulated GSH metabolism reported for high-producing CHO cells [44,45]. Thus, in highly producing CHO cells the GSH metabolism would not rate limit the interaction with the redox biosensor. In contrast, a positive correlation between response to an oxidant and the expression level of a related biosensor was previously reported for HEK-293 cells [15]. This suggests that, in HEK-293 cells, a higher intracellular content of rxYFP warrants a more efficient out competition of the biosensor with other endogenous targets for the Grx-mediated equilibration with the GSH/GSSG pool. Nonetheless, cell-type dependent differences in H_2_O_2_ membrane permeability [46] cannot be ruled out as an additional major factor responsible for the differential response of CHO-GM and HEK-293. In fact, the H_2_O_2_ diffusion coefficient has been shown to be a determinant in the effectiveness of HUVEC, IMR-90 and PC12 cells to respond, particularly, to low concentrations of the oxidant [47].

In order to overcome the kinetic limitations imposed by the availability of endogenous Grx, several fluorescent protein-based redox biosensors have been fused to Grx [48,49]. This strategy favored a rapid equilibration (from minutes to msec) of the biosensor with the GSH/GSSG pool and proved to overcome the cell-type or even cell-to-cell differences in Grx abundance and activity [22]. However, our several attempts to generate stable CHO-GM and HEK-293 cell lines expressing the Grx1-roGFP2 biosensor proved unsuccessful. One reason for this failure is that over-expression of Grx may affect critical cell functions during the selection process, since there are several reports using this biosensor on transient expression studies [18,22,49].

Compared to the ratiometric roGFP2 and roClover [18,23], rxYFP is an intensiometric biosensor [17,50]. One advantage of ratiometric over intensiometric biosensors is that they allow cancelling out variations in fluorescence intensity due to cell-to-cell variable expression or other non-redox (i.e., pH) or non-physiological factors (i.e., photobleaching, partial quenching, etc.). However, in clonal stable cell lines, such as those generated in our study, biosensor expression is overall highly uniform within the cell population. Furthermore, by performing the calibration of the intensiometric biosensor in control samples (i.e., treatment with a permeant reducing and oxidizing agent: DTT and H_2_O_2_ or diamide) the potential effects of factors interfering with fluorescence can be unmasked or cancelled.

In the cytosol of CHO-GM, the redox probe evidenced a minor but significant and progressive oxidation one day in advance the cell culture entered into stationary phase. Factors other than an impaired glucose catabolism or the depletion of serum protective or growth functions appear to be responsible for the shift in the GSH/GSSG reported by rxYFP in CHO-GM cells. This observation is in agreement with different studies reporting changes in the glutathione-related metabolism during the in vitro growth of mammalian cells. For instance, a metabolomic study performed in a recombinant antibody-producer CHO cell line highlighted a downward trend of key metabolites associated to energy, glutathione, and glycerophospholipid pathways at the exponential–stationary transition phase of a batch culture [39]. In this study, the intracellular level of GSH and GSSG declined progressively, suggesting an impaired biosynthesis of the tripeptide and/or that the cells were experiencing oxidative stress. Concomitantly, the content of extracellular GSSG increased indicating that cells secreted this metabolite, likely, to maintain a homeostatic GSH/GSSG ratio inside them. Another study showed a significant (2-fold) upregulation of five genes encoding for amino acid transporters (e.g., glutamate and glycine) associated to glutathione metabolism during the transition from exponential to stationary phase of CHO cells [51]. This suggests a remarkable growth phase-dependent demand for this important redox metabolite. In fact, we found that glutamate, a precursor of GSH, was almost depleted from the extracellular medium at the late exponential growth phase (day 3) of CHO-GM-rxYFP cells. 

Proline is a non-sulfur containing amino acid that was consumed from the culture medium when CHO-GM cells reached the stationary phase, and its intracellular accumulation has been shown to protect HEK-293 cells against oxidative stress [52]. However, it is well known that proline oxidation to glutamate via reactions catalyzed by two different mitochondrial dehydrogenases may be a source of ROS formation due to electron leakage from the electron transport chain [53,54,55]. Thus, if the last oxidative pathway prevails during the stationary phase to compensate for glutamate requirements for GSH biosynthesis, it will likely entail a concomitant increase in the intracellular ROS levels and, hence, in rxYFP oxidation. Although we did not measure the availability of extracellular cysteine, another study reported a significant and progressive decline in this amino acid from day 3 and onwards in a fed-batch culture of CHO cells [39].

The metabolic changes accompanying the cell growth profile of CHO-GM cells during the batch culture are compatible with a shift from a glycolytic- to a TCA-based energetic metabolism. Under physiological conditions, H_2_O_2_ is a metabolic byproduct of the mitochondrial energetic metabolism [56] that may contribute to the intracellular decrease in the GSH/GSSG ratio reported by the redox biosensor during this metabolic transition. A similar metabolic behavior was reported for a highly productive recombinant antibody-producing CHO cell line subjected to a fed-batch culture [57]. Under this culture condition, the metabolic reprogramming was preceded by a significant drop in the NADPH/NADP^+^ and GSH/GSSG ratios (day 3), which was ascribed to mitochondrial-derived ROS due to increased TCA flux. Thus, this study and our data confirm that the concerted metabolic reprogramming occurring in CHO cells in a batch culture is tightly linked to the cell thiol-redox metabolism.

In accordance with other studies performed in CHO cells [58], alanine and glycine were two amino acids that, along with the by-product ammonia, accumulated at significant levels during the batch culture of CHO-GM cells. Such effect was attributed to the consumption and intracellular conversion of asparagine into these amino acids and ammonia [58]. This reinforces our statement that the expression of the biosensor does not perturb cell metabolism.

GM-CSF-mediated signaling involves the transient production of ROS in neutrophils [38]. CHO cells lack receptors for this cytokine and addition of an excess of the recombinant GM-CSF to the culture medium of CHO-GM-rxYFP cells did not trigger biosensor oxidation. This led us to rule out a contribution of the secreted recombinant cytokine to the remarkable rxYFP oxidation occurring during stationary phase, when GM-CSF accumulated at high levels in the extracellular medium. However, GM-CSF contains two disulfide bridges that are essential for the cytokine activity [59], and its oxidative folding (formation of disulfide bridges) occurs in the endoplasmic reticulum (ER) *via* the action of protein disulfide isomerases and with a net yield of one GSSG (and two GSH consumed) for each protein disulfide formed [60]. The ER lacks GSSG reductase activity, therefore, recycling of GSH takes place in the cytosol. Thus, in addition to the metabolic changes highlighted above, it is tempting to speculate that production of GM-CSF disulfides represents an additional source of oxidative stress that is detected by rxYFP (i.e., decrease in the GSH/GSSG ratio) and becomes remarkable in the more restrictive metabolic conditions of the stationary growth phase.

Feeding strategies, media formulations, and fine-tuning of environmental parameters (pH, pO_2_, pCO_2_, osmolality, etc.), in which extracellular supplements or substrates are incorporated or limited, are often used to reduce the accumulation of cytotoxic metabolic byproducts, to optimize energetic metabolism and to control ROS production [4,61,62,63]. More demanding strategies include the genetic modification of cell lines to up- or downregulate specific pathways, the so called “cell engineering”, with the aim to increase energetic efficiency, viability, productivity, and product quality [64,65,66]. With respect to GSH metabolism, some studies conducted in CHO cells suggested a link between cell productivity and high intracellular GSH content [46,47]. However, overexpression of the catalytic subunit of γ-glutamyl cysteine synthetase (GCL, the bottleneck enzyme in GSH biosynthesis) in CHO cells, which led to an increased intracellular concentration of GSH, did not improve the production of a monoclonal antibody in transient expression assays [67]. A subsequent study of a monoclonal antibody-producing CHO cell line where its GSH content was down-modulated by limiting cysteine supply or by chemical inhibition of GCL has shown a severe impairment in the productivity of the heterologous protein [68]. The parallel proteomic analysis conducted in this study further revealed additional metabolic adaptations of the cells in response to GSH depletion that involved the downregulation of a subset of glutathione S-transferase enzymes, TCA-cycle enzymes, and lipid biosynthesis (cholesterol in particular), and the upregulation of several antioxidant enzymes. Given that accumulation of GSSG is considered an early signal of cell cytotoxicity and apoptosis [69], it is clear that the proteomic and protein production changes associated to a decline in GSH level or in the GSH/GSSG ratio tends to compensate them and, hence, to maintain a physiological redox *milieu*.

Clearly, the intracellular GSH/GSSG ratio acts as a redox rheostat that modulates several cellular processes beyond cell survival. Therefore, monitoring it in a non-invasive and online fashion, as shown here for the redox reporter CHO and HEK-293 cell lines, might prove highly useful for testing and implementing feeding and genetic strategies aimed to improve cell physiology and productivity in the field of a biomanufacturing process development campaign.

## 5. Conclusions

For the first time, stable redox reporter cell lines of biotechnological interest were generated and phenotypically characterized. As implemented in our study, the fluorescent protein redox biosensor rxYFP functions as a redox antenna that picks up the GSH/GSSG ratio signal inside the cell in a dynamic fashion according to the availability of signal transmitters (Grx) and interferents (ROS scavengers). Expression of rxYFP in the cytosol has no side effects on cell phenotype and proved very sensitive for detecting minor redox perturbations in the GSH/GSSG ratio caused by endogenous (metabolic) or exogenous (external H_2_O_2_ bolus) stimuli. Importantly, microscopy and flow cytometry analysis can be used in a complementary manner to obtain spatial resolution and statistical robustness in the fluorescence-based analysis of the redox reporter cell lines.

The redox reporter cell lines characterized here represent an excellent model study to address fundamental questions of thiol-redox metabolism and for the optimization of biotechnological bioprocessing.

## Figures and Tables

**Figure 1 sensors-22-01324-f001:**
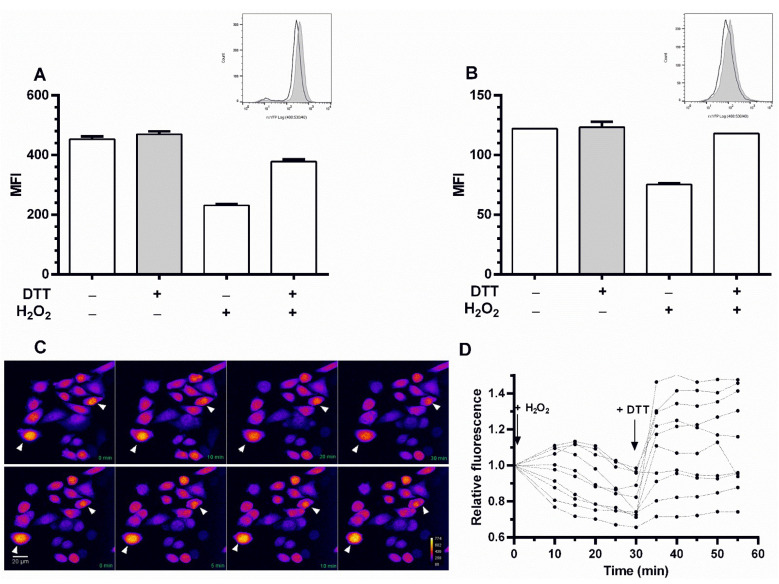
**Response of redox reporter cell lines to redox stimuli.** rxYFP MFI was measured in (**A**) CHO-GM-rxYFP and (**B**) HEK-293-rxYFP cells treated or not with 2.5 mM H_2_O_2_ (30 min) and/or with DTT (for CHO-GM-rxYFP: 1 mM for 20 min, and for HEK-293-rxYFP: 10 mM for 30 min). Data represent the mean of three measurements ± SD. The insets show representative rx-YFP fluorescence intensity histograms for non-treated (filled histogram) and H_2_O_2_-treated (2.5 mM for 30 min, empty histogram) cells. (**C**) Time-lapse measurement of rxYFP fluorescence by confocal microscopy. CHO-GM-rxYFP cells from non-synchronized cultures were treated with 2.5 mM H_2_O_2_ for 30 min (upper panel, images taken every 10 min starting from time 0) and then with 1 mM DTT for an additional 25 min (lower panel, images taken every 5 min starting from time 0). The images correspond to the sum of intensity projection for 20 Z-stacks and the fluorescence intensity is represented using a pseudo-color scale. The white arrows point to cells where the changes in fluorescence intensity are more pronounced. (**D**) Quantification of time-course redox-dependent changes in rxYFP fluorescence for 10 selected cells from panel (**C**). The relative fluorescence intensity of each cell is calculated as the MFI normalized to the cell area and the MFI at time 0.

**Figure 2 sensors-22-01324-f002:**
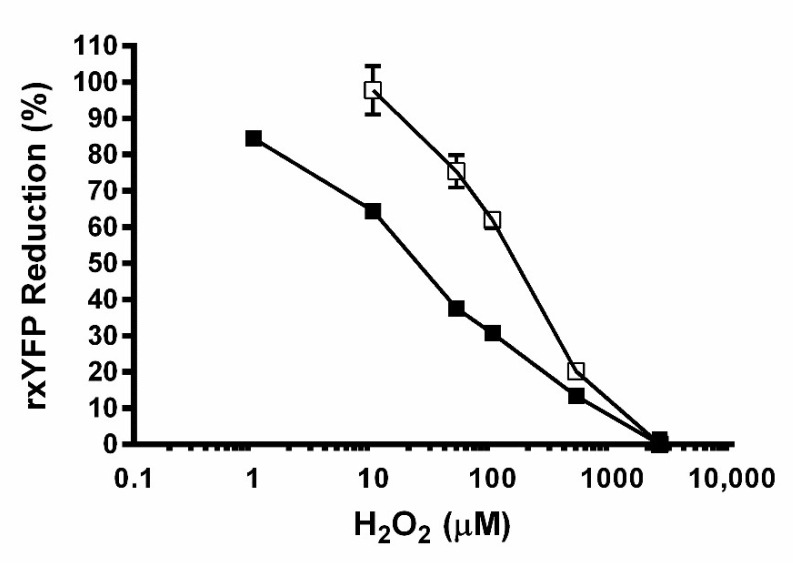
**Concentration-dependent response of redox reporter cell lines to H_2_O_2_.** CHO-GM-rxYFP (■) and HEK-293-rxYFP (□) cells were treated with different concentrations of H_2_O_2_ for 30 min and the MFI was evaluated by flow cytometry. The data shown correspond to a representative experiment out of two where the relative percentage of rxYFP reduction is expressed as the mean ± SD.

**Figure 3 sensors-22-01324-f003:**
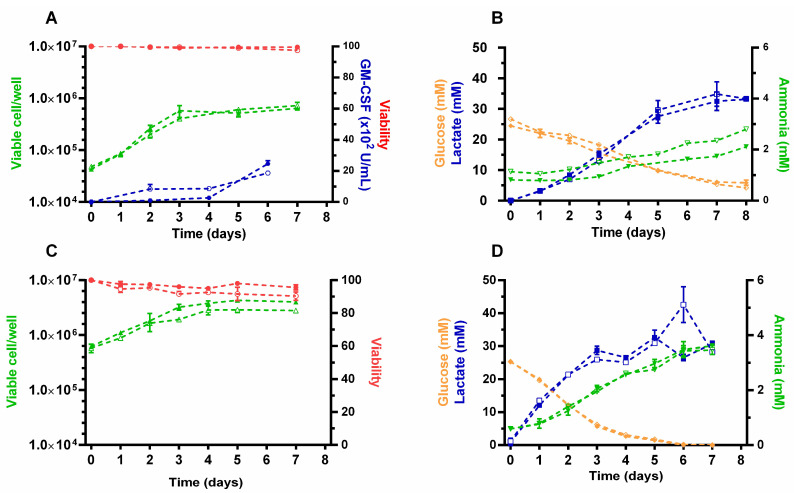
**Growth phenotype and metabolism of redox reporter cell lines.** Cell number (viable cells/well, △), cell viability (%,◦), recombinant hGM-CSF activity (U/mL, ⬡), glucose (mM, ◊), lactate (mM, □), and ammonia (mM, ▽) were determined at different time points of batch cultures from (**A**,**B**) CHO-K1-hGM-CSF and (**C**,**D**) HEK-293 cell lines, where the filled and empty symbols correspond to data from the parental and the redox reporter cells, respectively.

**Figure 4 sensors-22-01324-f004:**
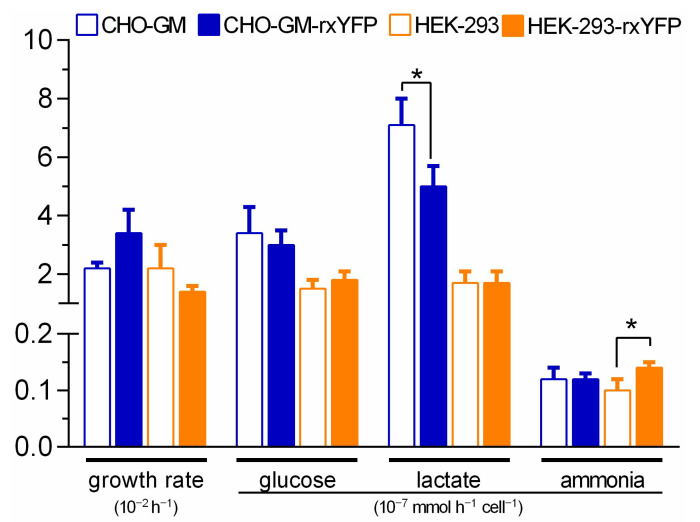
**Growth and metabolic rate of parental and redox reporter cell lines.** The specific growth rate, glucose consumption, and lactate or ammonia production during the exponential growth phase (day 0 to day 2) of parental (empty bars) and redox reporter (filled bars) cell lines is shown. Significant differences (*, *p* < 0.05) between reporter and parental cell lines from the same cell type are indicated with solid lines.

**Figure 5 sensors-22-01324-f005:**
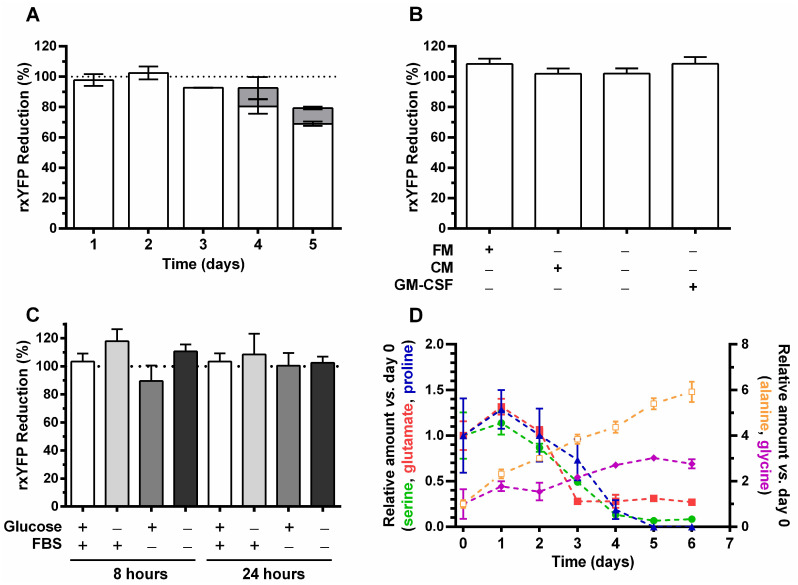
**Intracellular redox state of the biosensor during different culture conditions of CHO-GM-rxYFP cells**. (**A**) Intracellular level of reduced rxYFP (expressed as %) during a batch culture of CHO-GM-rxYFP. The grey bars at days 4 and 5 correspond to samples from cultures where the culture medium was fully replenished at days 3 and 4, respectively. (**B**) Effect on the intracellular redox state of rxYFP upon a 24 h treatment with conditioned medium or with rhGM-CSF (1 µg/mL). FM: fresh culture medium; CM: conditioned culture medium. (**C**) Effect on the intracellular redox state of rxYFP upon glucose or FBS deprivation for 8 or 24 h. The dashed line behind the bars indicates the 100% biosensor reduction. (**D**) Amino acid content (expressed relative to the concentration at day 0) in the supernatant of a batch culture of CHO-GM-rxYFP. Only amino acids that showed variations over time are plotted: serine (●), glutamate (▪), proline (▲), glycine (◆), and alanine (□). All data shown in the different experiments is expressed as the mean of three measurements ± SD.

## Data Availability

All data presented in this study are available upon request to the corresponding authors.

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
