# Peer review of "Generation and Characterization of Stable Redox-Reporter Mammalian Cell Lines of Biotechnological Relevance"

_sensors, 2022, doi:10.3390/s22041324_

Round 1
Reviewer 1 Report
Redox sensing in cells is increasingly important in light of recent advances in biomedical research, particularly in the fields of cancer and neurodegeneration. Use of fluorescent redox biosensors are proving to be valuable in examining pathogenesis and developing therapeutics. Therefore, this contribution from Perelmuter et al., which describes the generation and validation of stable cell lines expressing a redox-sensitive variant of YFP will be a welcome addition to the biomedical/biopharmaceutical literature. In this study, the authors ascertain that stable expression of rxYFP faithfully reflects changes in the oxidative state of cells without interfering with normal cell function. A wide array of processes that are expected to be sensitive to redox state are examined, and found to be unperturbed by expression of the probe. Along the way, the authors provide interesting insights into changes in the redox state of batch cultures with time in culture. I also wish to note the thoughtful and scholarly Discussion section and the clarity of the writing, although there are several misspellings, as listed below.
Regarding the data, I have only one comment: It is stated that there is some cell-to-cell variability in the fluorescence changes in response to H2O2 and DTT exposure. Unless I am misinterpreting Figure 1D, the variability seems quite large. Could the authors enlighten me on this issue?
Editing issues (misspellings, etc.):
Lines 152, 154, 350, 439, 546, 561, and 563: change “aminoacids” to “amino acids”.
Line 154: change “shortly” to “briefly”.
Line 373: change “lline” to “line”.
Line 391: change “to get some insight” to “of gaining insight”.
Line 422: (figure legend 5): change “replenish” to “replenished”.
Line 472: change “lover” to “lower” and “difussion” to “diffusion”.
Line 473: change “imply” to “implies”.
Line 485: change “similar to “similarly”.
Line 490: change “intracelllular” to “intracellular”.
Line 494: change “difussion” to “diffusion”.
Line 507: change “intesiometric” to “intensiometric”.
Line 514: change “teatement” to “treatment”.
Line 552: change “decreaset” to “decrease”.
Line 555: change “conditions” to “condition”.
Line 567: change “neutrophyles” to “neutrophils”.
Line 625: change “biotecnological” to “ biotechnological”.
Author Response
General comments from the authors:
We would like to thank the reviewers and editors of Sensors for their careful revision and assistance in improving our manuscript. Although not requested by the reviewers, the layout (colour, size) of some figures has been modified to improve their clarity. Below there is a point by point response to the reviewers’ concerns and observations in blue. We have also uploaded an updated manuscript containing the tracked changes and one with the changes accepted for your consideration.
Comments and Suggestions for Authors
Redox sensing in cells is increasingly important in light of recent advances in biomedical research, particularly in the fields of cancer and neurodegeneration. Use of fluorescent redox biosensors are proving to be valuable in examining pathogenesis and developing therapeutics. Therefore, this contribution from Perelmuter et al., which describes the generation and validation of stable cell lines expressing a redox-sensitive variant of YFP will be a welcome addition to the biomedical/biopharmaceutical literature. In this study, the authors ascertain that stable expression of rxYFP faithfully reflects changes in the oxidative state of cells without interfering with normal cell function. A wide array of processes that are expected to be sensitive to redox state are examined, and found to be unperturbed by expression of the probe. Along the way, the authors provide interesting insights into changes in the redox state of batch cultures with time in culture. I also wish to note the thoughtful and scholarly Discussion section and the clarity of the writing, although there are several misspellings, as listed below.
Regarding the data, I have only one comment: It is stated that there is some cell-to-cell variability in the fluorescence changes in response to H2O2 and DTT exposure. Unless I am misinterpreting Figure 1D, the variability seems quite large. Could the authors enlighten me on this issue?
Answer: The variability in the response of CHO-GM-rxYFP cells to different redox stimuli observed by fluorescence microscopy analysis (Fig. 1C-D) could be ascribed to the fact that this assay was performed with cells from non-synchronized cultures. Thus, the heterogeneous metabolic status of these cells may account for the different magnitude in rxYFP fluorescence upon redox stimulation. In addition, only a small number of cells from a randomly selected field was analyzed, which may not necessarily be representative of the whole cell population. On the other hand, this reinforces the more robust analytical and statistical power of a flow cytometry-based analysis, where thousands of cells are subject to interrogation.
Editing issues (misspellings, etc.):
Lines 152, 154, 350, 439, 546, 561, and 563: change “aminoacids” to “amino acids”.
Line 154: change “shortly” to “briefly”.
Line 373: change “lline” to “line”.
Line 391: change “to get some insight” to “of gaining insight”.
Line 422: (figure legend 5): change “replenish” to “replenished”.
Line 472: change “lover” to “lower” and “difussion” to “diffusion”.
Line 473: change “imply” to “implies”.
Line 485: change “similar to “similarly”.
Line 490: change “intracelllular” to “intracellular”.
Line 494: change “difussion” to “diffusion”.
Line 507: change “intesiometric” to “intensiometric”.
Line 514: change “teatement” to “treatment”.
Line 552: change “decreaset” to “decrease”.
Line 555: change “conditions” to “condition”.
Line 567: change “neutrophyles” to “neutrophils”.
Line 625: change “biotecnological” to “ biotechnological”.
Answer: All the misspelling and typographical mistakes listed above have been amended.
Reviewer 2 Report
The manuscript entitled Generation and characterization of stable redox-reporter mammalian cell lines of biotechnological relevance suggested the characterization of two transgenic cell lines that express stably in their cytosol a redox biosensor based on the yellow fluorescent protein, rxYFP. Historically, redox biology underwent great changes 2.3 billion years ago when living organisms had to adapt to a new life of the level of oxygen in Earth’s atmosphere. The most significant innovation in the history of life promoted the arrival of aerobic respiration and the occurrence of complex multicellular life. Redox characterization of system in cell is important issues. In general, fluorescent assay has contributed to advances in biochemistry that we now take for granted. Many biological and chemical processes of great importance in both nature and technology were uncovered using this versatile technique. Anyway, In this manuscript, the authors definitely elucidated stable redox reporter cell lines as a smart biosensor were generated and phenotypically characterized. I think it means an excellent model study to address fundamental questions of thiol-redox metabolism in biology. I think this is a useful article and I think it makes a potent scientific contribution in redox-reporter biosensor. I would recommend the manuscript for minor revision.
Just a minor point,
Introduction must be improved by incorporating more recent references including mammalian dominant system, bioprocess, redox system, redox system different fluorescent proteins and redox assays.
Author Response
General comments from the authors:
We would like to thank the reviewers and editors of Sensors for their careful revision and assistance in improving our manuscript. Although not requested by the reviewers, the layout (colour, size) of some figures has been modified to improve their clarity. Below there is a point by point response to the reviewers’ concerns and observations in blue. We have also uploaded an updated manuscript containing the tracked changes and one with the changes accepted for your consideration.
Comments and Suggestions for Authors
The manuscript entitled Generation and characterization of stable redox-reporter mammalian cell lines of biotechnological relevance suggested the characterization of two transgenic cell lines that express stably in their cytosol a redox biosensor based on the yellow fluorescent protein, rxYFP. Historically, redox biology underwent great changes 2.3 billion years ago when living organisms had to adapt to a new life of the level of oxygen in Earth’s atmosphere. The most significant innovation in the history of life promoted the arrival of aerobic respiration and the occurrence of complex multicellular life. Redox characterization of system in cell is important issues. In general, fluorescent assay has contributed to advances in biochemistry that we now take for granted. Many biological and chemical processes of great importance in both nature and technology were uncovered using this versatile technique. Anyway, In this manuscript, the authors definitely elucidated stable redox reporter cell lines as a smart biosensor were generated and phenotypically characterized. I think it means an excellent model study to address fundamental questions of thiol-redox metabolism in biology. I think this is a useful article and I think it makes a potent scientific contribution in redox-reporter biosensor. I would recommend the manuscript for minor revision.
Just a minor point,
Introduction must be improved by incorporating more recent references including mammalian dominant system, bioprocess, redox system, redox system different fluorescent proteins and redox assays.
Answer: New (and updated) references dealing with the topics indicated by the reviewer have been added to the Introduction.